# Research of the Algebraic Multigrid Method for Electron Optical Simulator

**DOI:** 10.3390/e24081133

**Published:** 2022-08-16

**Authors:** Zhi Wang, Quan Hu, Xiao-Fang Zhu, Bin Li, Yu-Lu Hu, Tao Huang, Zhong-Hai Yang, Liang Li

**Affiliations:** 1Vacuum Electronics National Laboratory, School of Physical Electronics, University of Electronic Science and Technology of China, Chengdu 610054, China; 2Shenzhen Institute for Advanced Study, University of Electronic Science and Technology of China, Shenzhen 518000, China; 3School of Mathematical Sciences, University of Electronic Science and Technology of China, Chengdu 611731, China

**Keywords:** FEM, algebraic multigrid, aggregation-based, preconditioning

## Abstract

At present, electron optical simulator (EOS) takes a long time to solve linear FEM systems. The algebraic multigrid preconditioned conjugate gradient (AMGPCG) method can improve the efficiency of solving systems. This paper is focused on the implementation of the AMGPCG method in EOS. The aggregation-based scheme, which uses two passes of a pairwise matching algorithm and the K-cyle scheme, is adopted in the aggregation-based algebraic multigrid method. Numerical experiments show the advantages and disadvantages of the AMG algorithm in peak memory and solving efficiency. The AMGPCG is more efficient than the iterative methods used in the past and only needs one coarsening when EOS computes the particle motion trajectory.

## 1. Introduction

At present, there is very little software dedicated to the simulation design of traveling wave tubes. Electron optical simulator (EOS) is 2D and 3D steady state beam trajectory software which is used to design traveling wave tubes [1]. EOS uses the finite element method (FEM) [2] for the solving partial differential equations (PDEs). The solution of the FEM linear system *Ax* = *b* arising from the FEM gives an approximation of the solution to PDEs. The FEM system of equations is large, with a sparse, symmetric and positive definite stiffness matrix *A*, with the stiffness matrix having no more than 20 non-zero elements per row even in the large FEM system. When EOS performs electromagnetic simulations of complex models, it needs to solve sparse linear systems, and its solution time accounts for more than half of the total simulation time. The solver of EOS needs to improve the solving speed, thereby improving the work efficiency of engineers.

For a large sparse system of linear equations *Ax* = *b*, the Krylov subspace methods (e.g., conjugate gradient (CG)) are preferred for their good convergence. Because the FEM systems of EOS are also ill-conditioned systems, the solution *x* to *Ax* = *b* is extremely sensitive to changes in A and b [3,4]. Therefore, the JPCG method of EOS fails to solve these ill-conditioned systems within a reasonable computer elapsed time. The algebraic multigrid (AMG) method is one of the most efficient solution techniques for solving linear systems arising from the discretization of second-order elliptic PDEs. COMSOL Multiphysics and ANSYS SpaceClaim both have AMG solvers to solve large-scale sparse linear systems, and the AMG method is often used as a preconditioner in Krylov subspace solvers [5,6]. Meanwhile, many scholars have proposed algebraic multigrid preconditioned conjugate gradient (AMGPCG) solvers [7,8,9,10], which are used to solve the problem of poor convergence in ill-conditioned linear systems. This paper uses the aggregation-based AMG as a preconditioner to solve the FEM systems.

This paper is organized as follows. Section 2 presents the principles and implementation details of the AMGPCG. Section 3 shows the computational efficiency of different iterative methods for different problems, and the most suitable scheme for EOS is finally determined. In Section 4, a simple conclusion is made for this paper.

## 2. Introduction and Implementation of the AMGPCG Method

To solve large sparse systems of linear equations quickly, the iterative method selected was the AMGPCG algorithm. It is necessary to study the specific algorithm theory and the implementation difficulties of the AMG, such as the time of stopping coarsening, the method of quickly solving the coarse grid matrix and the way of combining the AMG and CG methods.

### 2.1. AMG Method

The multigrid (MG) algorithm is one of the most effecient numerical methods for solving large-scale systems arising from (elliptic) PDEs. For large-scale FEM systems, the local relaxation methods (e.g., Gauss–Seidel) are typically effective at eliminating the high-frequency error components, while the low-frequency parts cannot be eliminated effectively [11]. The MG method’s idea is to project the error obtained after applying a few iterations of the local relaxation methods onto a coarser grid. The low-frequency part of the error on the find-grid becomes a relatively high-frequency part on the coarser grid, and these frequencies can be further corrected by a local relaxation method on the coarse grid. The MG method is repeating this process in ever coarser grids [12]. The local relaxation method is called smoother in this process.

The MG algorithm divides into the geometric multigrid (GMG) and algebraic multigrid (AMG) methods. The GMG method depends on a hierarchy of geometric grids. Since the geometric multi-grid is constructed by the geometric information, it is very difficult to generate a coarse grid for complex geometric structures. The AMG methods were put forward to solve the challenges of the multiple geometric algorithms [13]. They only need the coefficient matrix *A* of linear systems, which do not require different grid levels. A. Brandt, K. Stüben et al., proposed and developed the AMG method within the past three decades [14,15,16,17]. There are two kinds of methods that have been developed greatly. One method is constructing coarse matrixes and interpolation operators based on geometric parts and analytical information in linear systems, such as the smooth aggregation-based (SA-AMG) [18], energy-min AMG [19] and aggregation-based AMG methods [17]. The other is the adaptive AMG methods. Their basic idea is to adjust and optimize the AMG components in the solution process. This kind of method includes being based on methods such as compatible relaxation AMG [20], Bootstrap AMG [21] and root node-based AMG [16]. In this paper, the aggregation-based AMG proposed by Y. Notay [17] is used as the preprocessing condition of the AMGPCG method.

The AMG method consists of two phases: the setup phase and the solution phase [22]. The set-up phase first needs to design a multigrid $\Omega_{1}⊃\Omega_{2}⊃\ldots ⊃\Omega_{m}$. $\Omega_{k}\left(k=1,2,\ldots ,m\right)$ is set by the coarsening algorithm. $\Omega_{m}$ is the top-level grid. $\Omega_{k}=C_{k}\cup F_{k}$, $C_{k}$ is a set of coarse grid points, and $F_{k}$ is a set of thin grid points. $C_{k}$ and $F_{k}$ do not intersect. Then, the set-up phase needs to construct a grid operator $A_{c}$ and interpolator *P* for every $\Omega_{k}$. The solution phase performs multigrid loops, such as V-cycle, W-cycle and FMG.

The coarsening algorithm can construct $P^{n\times n_{c}}$ using only the information of $A^{n\times n}$. The coarse grid matrix ${A_{c}}^{n_{c}\times n_{c}}$ is computed by the Galerkin formula:(1)$$A_{c}=P^{T}AP.$$

The algebraic coarsening algorithm has classical coarsening and aggregate roughing, among others. Aggregation coarsening is used in this paper. The coarse grid point set defines the aggregates $G_{i}$, and the interpolation matrix *P* is constructed from $G_{i}$ as follows:   
(2)$$P_{ij}=\left\{\begin{array}{c} 1,if\;i\in G_{j}\\ 0,otherwise \end{array}\right.\left(1\leq i\leq n,1\leq j\leq n_{c}\right).$$

$G_{i}$ requires the set of nodes $S_{i}$ to which *i* is strongly negatively coupled using the strong or weak coupling threshold β:(3)$$S_{i}=\left\{j\neq i|\;a_{ij}<-\beta {max}_{a_{ik}<0}|a_{ik}|\right\},$$
where β=0.25. EOS constructs a finite element matrix in which Dirichlet boundary conditions have been imposed. $m_{i}$ is the number of unmarked nodes that are strongly negatively coupled to *i* ($m_{i}$ is the number of sets $S_{j}$ to which *i* belongs and that correspond to an unmarked node *j*):(4)$$m_{i}=\left|\left\{j|i\in S_{j}\right\}\right|.$$

Algorithm 1 is part of the coarsening [17], and it finds coarse grid aggregations $G_{i}$, where $i=1\ldots n_{c}$.
**Algorithm 1** Pairwise aggregation.**Input:** Matrix $A^{n\times n}\;$;  Sets $\;S_{i}\;,\;i=1\ldots n$;Sets $m_{i}\;,\;i=1\ldots n$;Array $U=\left[1,n\right]$;The bool (whether the grid $\Omega_{k}$ is top-level grid $\Omega_{m}$) chose;The number of coarse variables $n_{c}=0$;**Output:** aggregation $G_{i}\;,\;i=1\ldots n_{c}$.1:**if** chose==true **then**2:   *U*\$\left\{i|a_{ii}>5\sum_{i\neq k}|a_{ik}|\right\}$;3:**end if**4:**while** U≠
 **do**5:    Select i∈U with minimal $m_{i}$; $n_{c}$= $n_{c}$+1;6:    Select j∈U such that $a_{ij}=min_{k\in U}a_{ik}$;7:   **if** $j\in S_{i}$ **then**8:     $G_{n_{c}}=i,j$;9:   **else**10:     $G_{n_{c}}=i.$;11:   **end if**12:   U=U\$G_{n_{c}}$;13:   For all $i\in G_{n_{c}}$, updata: $m_{l}=m_{l}-1$ for $l\in S_{i}$;14:**end while**

The technical difficulty of Algorithm 1 is to select i∈U with minimal $m_{i}$. Time complexity in the bubble sort is $O\left(m\right)$, so it cannot quickly find the minimal $m_{i}$ by sorting $m_{i}$. This paper uses minimum heap sorting to quickly select i∈U with a minimal $m_{i}$. The minimum heap data structure is a complete binary tree. This complete binary tree has nodes whose values are less than their children, as shown in Figure 1. The minimum heap in Figure 1 shows the data structure of a binary tree. Because it is too complex to store the minimum heap data in a full binary tree, the minimum heap in Figure 1 stores it in an array, as shown in Figure 2.

We can use step 5 of Algorithm 1 to take out the root node in the min-heap. Step 13 of Algorithm 1 updates $m_{j}$, and the order of the heap is restored through the upper filtering of the min-heap. This paper uses an array Gn to store $G_{i}\left(1\leq i\leq n_{c}\right)$. If $G_{i}$ has two points, then $G_{i}=j,k$, Gn inserts nodes *j* and *k*. If there is only one point, then $G_{i}=j$, Gn inserts *j* and −1. $G_{i}$ can be used by the array index.

Algorithm 1 achieves aggregation, which is always in pairs. However, coarsening by pairwise aggregation is slow. Repeating double pairwise aggregation can result in faster coarsening. The aggregation ${G_{i}}^{\left(1\right)}\;\left(1\leq i\leq n_{c1}\right)$ is constructed from matrix *A* based on Algorithm 1. We construct auxiliary matrix $A_{1}$ as follows:(5)$$({A_{1})}_{ij}=\sum_{k\;\in \;{G_{i}}^{\left(1\right)}\;}\sum_{l\;\in {G_{i}}^{\left(1\right)}\;}a_{kl}\left(1\leq i,j\leq n_{c1}\right).$$

The aggregation ${G_{i}}^{\left(2\right)}\;\left(1\leq i\leq n_{c2}\right)$ is constructed by $A_{1}$ based on Algorithm 1. Then, we construct an aggregation $G_{i}\left(1\leq i\leq n_{c}=n_{c2}\right)$, which is given by
(6)$$G_{i}\;=\cup_{{j{\in G}_{i}}^{\left(1\right)}}{G_{j}}^{\left(2\right)}.$$

These aggregates $G_{i}$ are mostly quadruplets, with some triplets, pairs and singletons left. Aggregations $G_{i}\left(1\leq i\leq n_{c}\right)$ can construct the coarse matrix ${A_{c}}^{n_{c}\times n_{c}}$ and the interpolator $P^{n\times n_{c}}$ with Equations (Equation 1) and (Equation 2):(7)$${\left(A_{c}\right)}_{ij}=\sum_{k\;\in G_{i}\;}\sum_{l\;\in G_{j}\;}{\left(A\right)}_{kl}\left(1\leq i,j\leq n_{c}\right).$$

Because the nonzero entry values of the interpolator *P* are all one, and there is only one nonzero entry in one row, then the coefficient matrix *A* and the interpolator *P* are sparse matrixes. In accumulation $\sum_{k\;\in G_{i}\;}\sum_{l\;\in G_{j}\;}{\left(A\right)}_{kl}$, many cumulative points ${\left(A\right)}_{kl}$ are zero. Therefore, accumulation $\sum_{k\;\in G_{i}\;}\sum_{l\;\in G_{j}\;}{\left(A\right)}_{kl}$ only computes when $\left(G_{i}\;\cup G_{j}\;\right)\neq \emptyset$.

With the multigrid $\Omega_{1}⊃\Omega_{2}⊃\ldots ⊃\Omega_{m}$, the common coarsening stopping condition is that the order of the coarse matrix $A_{c}$ is less than 100. However, coarsening is very slow for the individual multigrid. An example is listed in Table 1, in which $\Omega_{k}$ is the grid layer and *n* is the matrix order number of the grid layer. If k>178, then the coarsening is very fast. If k≤178, then the coarsening effect is feeble. The whole coarsening efficiency is poor. According to this phenomenon, $\frac{n}{n_{c}}<x$ is used as the coarsening stopping index, and we finally found that x=1.25 could meet our requirement. Therefore, if $n_{c}<100$ or $\frac{n}{n_{c}}<1.25$ is met during coarsening, then coarsening is stopped.

After constructing the coarse grid matrix $A_{c}$ and the interpolator operator *P*, the set-up phase of the AMG algorithm has been implemented. The solution phase performs two schemes. In EOS, the V-cycle and K-cycle schemes are both tested.

The AMG program in EOS can be implemented in parallel theoretically. The CPU transfers the find-grid matrix *A* from the CPU’s DRAM to the GPU’s device memory. Then, the AMG finds coarse grid aggregations $G_{i}$ on the GPU by a mutex lock. Equation (Equation 1) proves that the solutions of the coarse matrix element ${A_{c}}_{ij}$ do not affect each other’s elements. Therefore, the program of computing ${A_{c}}_{ij}$ can be evaluated in parallel on the GPU, and the GPU can be used to compute intensive smoothing operations at the same time [7].

### 2.2. The AMGPCG Method

In EOS, the large FEM linear systems Ax=b are ill-conditioned, as the matrix *A* has large condition numbers $\kappa \left(A\right)=∥A∥∥A^{-1}∥$. Therefore, the JPCG and CG methods in EOS fail to converge within a reasonable computer elapsed time. The number of iterations in the Krylov subspace methods is reduced drastically when the AMG method is used as a preconditioner. The CG method and bi-conjugate gradient stabilized method are Krylov subspace methods. The CG method requires the coefficient matrix to be symmetric and positively definite. The bi-conjugate gradient stabilized method is a Krylov method which relaxes the symmetric constraint. The FEM linear systems satisfy the conditions of the CG method. Therefore, this paper uses the AMGPCG method to solve the FEM linear systems in EOS.

## 3. Comparison of Algorithms

This section focuses on the computational efficiency of different iterative methods for different finite element systems in the systems of linear equations generated by EOS, and the efficiencies of the K-cycle and the V-cycle are also compared. For this, four systems of linear equations were generated with the same computational model and different minimal grid sizes, which are denoted as P1–P4. All the simulations were performed on a workstation configured with an Intel(R) Xeon(R) Gold 6240 CPU @ 2.60 GHz with a memory size of 16 GB.

### 3.1. Verify the Availability of the AMGPCG Solver

In addition to the AMGPCG algorithm program, the AMGPCG solver also has a sparse matrix operation program. The sparse matrix operation program needs to use appropriate sparse matrix storage. We analyzed the ternary storage, cross-linked list storage and row compression storage (CSR). Because the stored coefficient matrix is often multiplying, this paper selected the CSR method as the sparse matrix storage scheme in EOS. Then, the sparse matrix operations based on the CSR storage method were implemented: matrix sequential storage, matrix transpose, matrix addition and subtraction, matrix multiply vector, matrix multiply matrix, matrix row transformation and so on.

After completing the AMGPCG solver, one needs to test the availability of the AMGPCG solver. This paper tested a case in which an electron gun was computed by the AMGPCG solver and JPCG solver. This electron gun was divided into 312,953 grid faces and 157,902 grid points, and the order of the matrix was 628,775. The electron trajectories were the same as those constructed by the two solvers in Figure 3. In Figure 4, the convergence curves were convergent with the AMGPCG solver and JPCG solver. The calculation results of the AMGPCG solver and JPCG solver were the same. As such, the total cathode emission currents were both 269.1840 mA. The injection waist radii were both 0.6213 mm. The injection waist positions were both 9.27 mm. The results of the AMGPCG solver and JPCG solver were the same, so our AMGPCG solver was available.

### 3.2. The Information of the Coefficient Matrix

The key information of the coefficient matrix in the four systems of linear equations were provided and analyzed. In the finite element systems *Ax* = *b*, the initial array $x^{0}$ elements were all zero. For convergence, the elements in the error array $r=b-Ax^{*}$ must be less than $10^{-10}$. Table 2 lists the key information of the coefficient matrix $A^{n*n}$ in *Ax* = *b* in EOS. Four cases with different minimal grid sizes are provided and denoted under “Problem” in Table 2, where “*n*” is the order of the matrix $A^{n*n}$. In “$\frac{nnz\left(A\right)}{n^{2}}$”, “$nnz\left(A\right)$” stands for the number of nonzero elements, and “%PosD” is the percentage of positive diagonal elements. In all the cases, $\frac{nnz\left(A\right)}{n^{2}}<10^{-3}$, and the larger the order of the matrix is, the smaller $\frac{nnz\left(A\right)}{n^{2}}$ will be, while %PosD = 100% shows that the diagonal is positive.

### 3.3. Comparison of the Calculation Effects

The K-cycle in the AMGPCG method was developed by Y. Notay [17]. Table 3 lists the calculation times and the iterations of the V-cycle and K-cycle with *t* = 0.00 and *t* = 0.25 for the four systems of linear equations. As can be seen in Table 4, no matter how big or small the matrix was, the K-cycle with *t* = 0.25 was still faster than the K-cycle with *t* = 0 and the V-cycle. In the K-cycle, the set of *t* = 0 indicates two iterations in each grid layer, and *t* = 0.25 indicates the possibility to iterate only once. Therefore, the higher the number of grid layers, the more pronounced the performance of the K-cycle with *t* = 0.25. This can explain the advantage of the K-cycle with *t* = 0.25 to some extent.

Table 4 compares the calculation times and the iterations for the AMGPCG, JPCG, CG and Gauss–Seidel methods for the four systems of linear equations. In the AMGPCG method, the K-cycle with *t* = 0.25 was used. It is easy to see that with a larger matrix scale, the AMGPCG method showed better performance than the other three algorithms in terms of the iteration number and solution time. For large sparse linear systems, the AMGPCG method was 17 times faster than the CG method and 6.5 times faster than the originally used JPCG method. However, the AMGPCG algorithm sacrificed memory usage, as the peak memory occupied by the AMGPCG solution was about twice that of the JPCG method, as shown in Figure 5, and the AMGPCG method could improve the solving speed by paralleling the AMG program [7].

The set-up phase consumed a large proportion of the AMGPCG algorithm’s time. Table 5 lists the time proportion of the set-up phase for the full AMGPCG iteration for the four linearsystems (P1–P4). All linear systems have $\frac{setup}{total}>50\%$ in Table 5, and because the particle trajectory computation was solving *Ax* = *b* with different *b* and the same *A*, the set-up phase only needed to be executed once, leading to higher efficiency for the AMGPCG method compared with other iterators for particle trajectory computation. Figure 6 provides the calculation times for each of the trajectory computations *Ax* = *b* for cases P1–P4. The AMGPCG method needed a set-up phase on the first trajectory solution *Ax* = *b*, so the solving time was long. However, after the first trajectory solution, the later trajectory solution *Ax* = *b* only changed the array *b* and did not change matrix *A*. Therefore, the later trajectory solution *Ax* = *b* only needed the solution phase, which caused the later trajectory solution *Ax* = *b* to go very fast. From Table 5 and Figure 6, we know it is extremely efficient to use AMGPCG as the solution method to compute the particle trajectory.

## 4. Conclusions

This paper presented an AMGPCG method for EOS. It works for solving systems of linear equations. The efficiency of the K-cycle scheme in the AMGPCG method was better than the V-cycle scheme in testing. The AMGPCG, JPCG, CG and Gauss–Seidel methods had their solution time and calculation iterations compared. It was proven that the AMGPCG method was faster and more robust for large linear systems. Currently, there is no parallel for the AMGCG method. When constructing a coarse matrix $A_{c}$, ${{(A}_{c})}_{ij}$ was not affected by the other points because of Equations (Equation 1) and (Equation 2). Therefore, the set-up phase can involve parallel computing. The multi-line parallel construction will significantly improve the calculation speed in the set-up phase. 

## Figures and Tables

**Figure 1 entropy-24-01133-f001:**
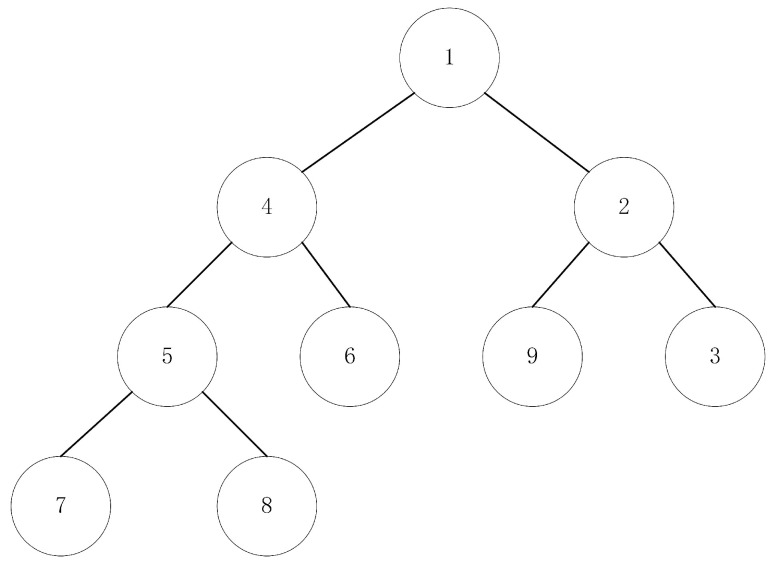
Minimum heap model.

**Figure 2 entropy-24-01133-f002:**
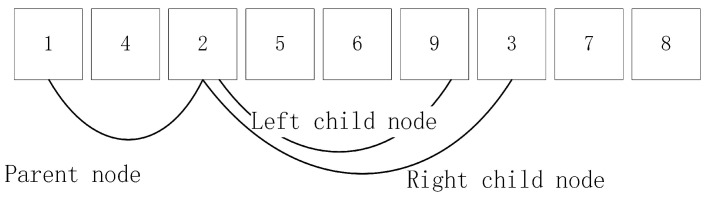
The array model of the smallest heap.

**Figure 3 entropy-24-01133-f003:**
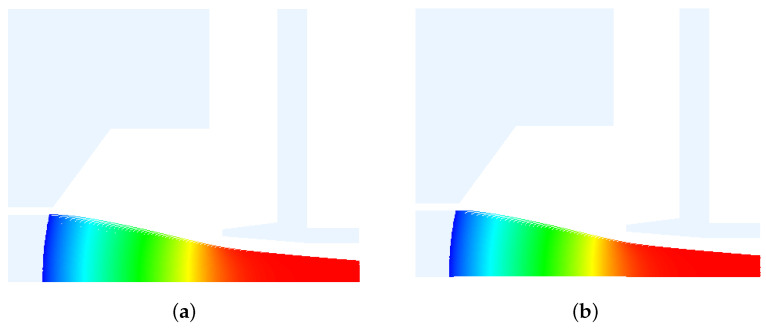
Electron trajectories of the AMGPCG solver and JPCG solver in EOS. (**a**) Electron trajectory with the AMGPCG solver. (**b**) Electron trajectory with the JPCG solver.

**Figure 4 entropy-24-01133-f004:**
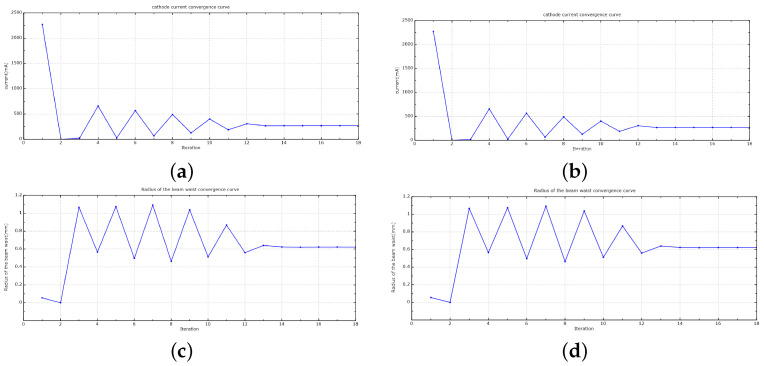
Convergence curves of the AMGPCG solver and JPCG solver in EOS. (**a**) Cathode current convergence curve with the AMGPCG solver. (**b**) Cathode current convergence curve with the JPCG solver. (**c**) Radius of the beam waist convergence curve with the AMGPCG solver. (**d**) Radius of the beam waist convergence curve with the JPCG solver.

**Figure 5 entropy-24-01133-f005:**
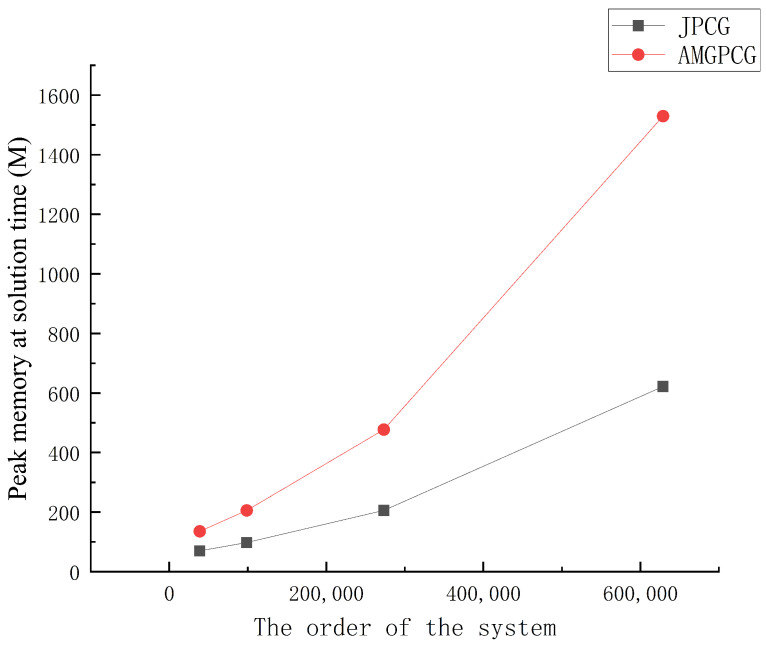
Peak memory occupied by the AMGPCG and JPCG methods.

**Figure 6 entropy-24-01133-f006:**
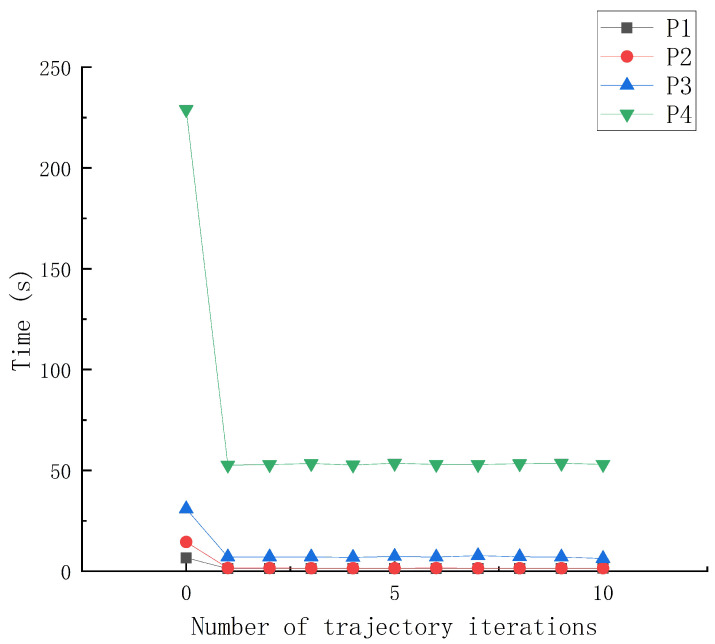
Computation time for every trajectory iteration by the AMGPCG method.

**Table 1 entropy-24-01133-t001:** The multigrid contains the order of the matrix.

$\Omega_{k}$	$\Omega_{188}$	$\Omega_{186}$	$\Omega_{182}$	$\Omega_{180}$	$\Omega_{178}$	$\Omega_{139}$	$\Omega_{89}$	$\Omega_{1}$
*n*	1,220,147	96,788	1488	820	694	477	377	201

**Table 2 entropy-24-01133-t002:** Key information of coefficient matrix $A^{n*n}$ in EOS.

Problem	*n*	$\frac{\mathrm{nnz}\left(A\right)}{n^{2}}$	%PosD
P1	47,581	0.00023	100%
P2	75,755	0.00015	100%
P3	191,714	0.00005.9	100%
P4	1,220,147	0.0000094	100%

**Table 3 entropy-24-01133-t003:** K-cycle and V-cycle.

Problem	Time(s)	Iterations
	Problem: P1	
V-cycle	24	82
K-cycle *t* = 0.00	29	22
K-cycle *t* = 0.25	7	23
	Problem: P2	
V-cycle	60	100
K-cycle *t* = 0.00	76	22
K-cycle *t* = 0.25	14	23
	Problem: P3	
V-cycle	268	178
K-cycle *t* = 0.00	79	23
K-cycle *t* = 0.25	30	23
	Problem: P4	
V-cycle	641	181
K-cycle *t* = 0.00	519	23
K-cycle *t* = 0.25	229	23

**Table 4 entropy-24-01133-t004:** Comparison of different iterative methods.

Problem	Time(s)	Iterations
	Problem: P1	
AMGPCG	7	23
JPCG	6	1297
CG	13	3500
Gauss–Seidel	239	63,450
	Problem: P2	
AMGPCG	14	23
JPCG	13	1642
CG	32	4423
Gauss–Seidel	508	100,791
	Problem: P3	
AMGPCG	30	23
JPCG	66	2680
CG	158	7258
Gauss–Seidel	4269	242,406
	Problem: P4	
AMGPCG	229	23
JPCG	1500	6771
CG	3891	17,875
Gauss–Seidel	–	>1,000,000

**Table 5 entropy-24-01133-t005:** The set-up phase accounting for the total calculation.

Problem	P1	P2	P3	P4
$\frac{setup}{total}$	80.75%	89.09%	77.18%	77.02%

## Data Availability

The theoretical data generated by cellular automata simulations is available from the authors upon a reasonable request.

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
