# Peer review of "Research of the Algebraic Multigrid Method for Electron Optical Simulator"

_entropy, 2022, doi:10.3390/e24081133_

Round 1
Reviewer 1 Report
The authors present a simple AMG method for EOS. The provide the algorithm details and show some numerical results.
The proposed idea is nice, but too simple. Currently available multigrid solvers are much more evolved than what proposed.
If the authors want to present their method, they must to:
- describe the matrix properties (is it symmetric?, which is its kernel?, ...)
- specify if they target a classical or aggregation-based multigrid
- compare they choices with what present in recent literature
- use much more updated references. Right now, the most recent has at least 10 years
- write a better introduction, describing the main multigrid algorithm and collocate their among what already present
- compare the performances with other available MG software. To beat Jacobi is quite trivial
- deal with parallel implementation issues (at least theoretically)
Minor comments are:
- revise the English
- define all symbols. E.g., introduction of Section 2: what is m_i?
- avoid * to mean multiplication among matrices
- use x instead * to mean high order dimensions, e.g., n x m, instead of n * m_i
- fig. 3: usually condition numbers grow up to 10^15 without any issues. They come after 1/machine precision, thus 40000000 is a small number.
- please, avoid PCG algorithm. It can be found in any textbook and even in Wikipedia
- provide convergence profiles in terms of residual vs iterations
- better describe the chosen problems and fairly compare them with other MG
Reviewer 2 Report
The paper tackles the algebraic multigrid preconditioned conjugate gradient method to enhance the efficiency of the Electron Gun and Collector Modeling Code [1].
Numerical results show that the EOS solver phase can greatly benefit from the use of the AMGPCG preconditioner.
The use of Krylov subspace methods accelerated with algebraic multigrid preconditioning are of standard use in today’s more efficient solvers. The paper is well organized, but English needs a deep revision. Section 2, deploys well-known information and deserves a better and clear description. Some good numerical tests are performed and posted in Section 3, but the English is so weak and the writing careless, which makes reading difficult and devalues the work. The paper must be carefully edited by a proof-reader. The novelty of the paper is limited.
[1]. Hu, Q.; Huang, T.; Yang, Z.H.; Li, J.Q.; Jin, X.L.; Zhu, X.F.; Hu, Y.L.; Xu, L.; Li, B. Recent Developments on EOS 2-D/3-D Electron 195 Gun and Collector Modeling Code. IEEE Transactions on Electron Devices (2010), 57, 1696–1701.
1) The acronym EOS must be explicitly indicated in the abstract. The code is not so well known for the reader to understand what the authors are referring to. This also applies to using EOS as a keyword.
2) What the authors refer to when they mention “coaching”? (line 41)
3) In line 121, section 3, the authors write “…I analyze the ternary storage …”, This is a work of several authors?
4) Better to state “Y. Notay [11]” instead of “paper of professor YVAN NOTA [11]”
Minor comments: (just for pages 1 and 2)
Lines 18-20: The authors may want to consider “Therefore, the efficiency of EOS needs to be improved by using a better preconditioner than the usual traditional Jacobi preconditioned conjugate gradient (JPCG) method. A better solver can speed up the solution of electromagnetic problems, thereby improving the work efficiency of engineers.”
Line 29: “system” instead of “equation”
Line 35: “Section” instead of “Sections”
Line 45: “numerical method” instead of “numerical solutions”
Line 47: the author may want to consider “… a smoother”
Line 49; Maybe “reduced” instead of “minor”
Line 57; “… proposed …”
Round 2
Reviewer 1 Report
The authors addressed all the raised issues. Now the work can be published.
Reviewer 2 Report
The authors carried out most of the recommendations proposed to them. The work, in my opinion, is now better.